# Origins of Life Research: The Conundrum between Laboratory and Field Simulations of Messy Environments

**DOI:** 10.3390/life12091429

**Published:** 2022-09-14

**Authors:** David Deamer

**Affiliations:** The Department of Biomolecular Engineering, University of California, Santa Cruz, CA 95064, USA; deamer@soe.ucsc.edu

**Keywords:** origin of life, messy conditions, laboratory simulations

## Abstract

Most experimental results that guide research related to the origin of life are from laboratory simulations of the early Earth conditions. In the laboratory, emphasis is placed on the purity of reagents and carefully controlled conditions, so there is a natural tendency to reject impurities and lack of control. However, life did not originate in laboratory conditions; therefore, we should take into consideration multiple factors that are likely to have contributed to the environmental complexity of the early Earth. This essay describes eight physical and biophysical factors that spontaneously resolve aqueous dispersions of ionic and organic solutes mixed with mineral particles and thereby promote specific chemical reactions required for life to begin.

## 1. Introduction

Virtually all experimental results that guide research related to the origin of life are from laboratory simulations of the early Earth conditions. How good are the simulations, and how confident can we be that the results are applicable to understanding the process by which life began 4 billion years ago in chemically complex and messy environments? One way to think about this question is to ask what we mean by “messy.” In the laboratory, emphasis is placed on the purity of reagents and carefully controlled conditions, so there is a natural tendency to reject impurities and lack of control. The reason is obvious: scientists want their results to be repeatable by others. 

Another meaning of messiness refers to environments that incorporate multiple biological and non-biological compounds [1,2,3]. An example is Bumpass Hell, a hydrothermal site on the Mount Lassen volcano in northern California (Figure 1). The image shows extensive deposits of clay, a common mineral in hydrothermal pools associated with volcanism. The pool is acidic, with pH 3, due to sulfur dioxide dissolved in the water, and the temperature is maintained at 91 °C by residual heat from the underlying magma that powered the eruption. It seems improbable that life could begin in such conditions, which, yet, are ubiquitous in volcanic regions of today’s Earth and presumably on the prebiotic Earth 4 billion years ago.

That brings us to the question being addressed in this essay: How could orderly structures and functions emerge that led to the first forms of life? The answer must involve physical and chemical processes that occur even in messy conditions. We will refer to the result of such processes as resolution, meaning that relatively pure compounds are produced and then concentrated in such a way that they can react despite the complexity of their surroundings. However, it is helpful to realize that messiness can be divided into chemical and physical components, and that there are synergistic combinations in which certain physical conditions can promote specific chemical reactions. Each of the following sub-topics are examples of such interactions.

## 2. Physical Self-Assembly Processes Tend to Resolve Complex Mixtures

A number of concerns arise when a chemist considers a natural environment in comparison to a laboratory one. Some of these are related to the composition of a mixture, while others come from physical variations such as temperature, pH, ionic solutes, adsorption to mineral surfaces and concentration, all of which are controlled in the laboratory. Let us begin with a few examples of such concerns:Desired reactions cannot occur if a solution is composed of so many solutes that the reactants are diluted.Side reactions interfere with yields if other solutes are present as potential reactants.If the temperature is too low, there will be insufficient activation energy, while too high temperatures will result in what a chemist refers to as tar [4,5]. Rather than controlled condensation reactions that synthesize specific bonds such as esters and peptides, elevated temperatures and high activation energy drive the formation of multiple, random bonds that produce the intractable polymers called asphalt and tar.Extreme pH ranges markedly affect the kinds of reactions that can occur and their yields.High salt concentrations, particularly, of divalent cations such as calcium, precipitate anions that are essential for life. Calcium, for instance, reacts with phosphate to form the insoluble mineral apatite.

How can we address these concerns? The approach described here is to realize that certain physical properties of complex mixtures can add order to a disorderly system, thereby promoting specific chemical reactions. For example, hydrothermal vents have been proposed as a site conducive to the origin of life [6], and the hot seawater flowing through alkaline vents is a highly complex mixture with available free energy and reducing power [7,8]. The effluent is produced by a serpentinization reaction, and ionic solutes precipitate as mineral crystals when the hot fluid encounters the cooler seawater bathing the vents. The minerals are mostly calcium carbonate, which illustrates an important point. Ionic compounds in a solution are disordered and in a high entropy state, but when a hot, alkaline solution is mixed with cool seawater the solubility limits of calcium and carbonate are exceeded, and crystallization begins. Crystals are the most ordered state of ionic solutes; therefore, simply mixing a vent fluid with seawater leads to a vast reduction of the entropy of the system when relatively pure calcium carbonate crystals precipitate.

## 3. Three Conditions Are Key to Resolving Messy Reaction Mixtures

Here, we will focus on three conditions that are ubiquitous in the laboratory but seldom invoked as solutions to deal with messy conditions on the early Earth. The conditions are related to concentration, water activity and temperature. For instance, when organic chemists synthesize an ester, they will mix pure, highly concentrated solutions of organic acids and alcohols rather than dilute solutions in an aqueous phase. The obvious reason is that water is a product of the condensation reaction that forms ester bonds, so the highest yields require that the water activity is low. The chemist will also heat the reaction mixture to provide activation energy that increases the rate of the reaction.

An important question is how linking bonds could be synthesized on the prebiotic Earth to provide the more complex molecules required for life to begin. Examples that have been explored in the laboratory include polyesters [9], peptides [10] and depsipeptides [11]. Taking a lesson from organic chemistry, the potential reactants must be concentrated, dry and at an elevated temperature. There is only one natural environment that meets these requirements: hydrothermal pools on subaerial volcanic land masses [12]. Precipitation provides fresh water distilled from a salty ocean, and the water in hydrothermal pools undergoes continuous cycles of evaporation and rewetting. If organic solutes are in the water, they will become extremely concentrated films on mineral surfaces during the dry phase of a cycle and then redissolve in the wet phase. Finally, the temperature is elevated, but typically less than the 100 °C boiling point of water because of the lower atmospheric pressure at higher altitudes. An important point is that even though a variety of solutes might be present in a messy solution, only those solutes capable of self-assembly or forming linking bonds will serve as reactants for polymerization, while the other solutes will be inert. Given the significance of concentration, we can now consider processes that can increase the local concentration of potential reactants.

## 4. Polar vs. Non-Polar: Oils and Monolayers at the Air–Water Interface

We tend to focus on water-soluble polar and ionized solutes when we think about constituents of messy mixtures, but we should also consider non-polar compounds such as aliphatic and aromatic hydrocarbons and their derivatives. One of the simplest resolutions in a messy environment is the fact that oils are relatively insoluble in water and are also less dense. Hydrocarbon oils therefore tend to separate spontaneously from other solutes and float on water surfaces. Furthermore, some of the hydrocarbons present in unrefined oil are likely to be amphiphilic compounds that have both polar and non-polar moieties in their structure. Amphiphiles are classified as surfactants because they spontaneously accumulate at air–water interfaces as monomolecular films and reduce the surface tension as a result.

Fatty acids are common amphiphiles that take their name from the fact that biological fat is a triglyceride with three fatty acids attached to a glycerol molecule by ester bonds. When the fatty acids are released from fat by hydrolysis of the ester bonds in alkaline solutions, the result is a natural detergent called soap. Palmitic acid is a 16carbon non-polar hydrocarbon chain with the terminal carbon in the form of a hydrophilic carboxyl group:CH_3_(CH_2_)_14_-COOH ⇌ CH_3_(CH_2_)_14_-COO^−^ + H^+^(1)

It is an organic acid because the terminal carboxy group can either be protonated (-COOH) or become anionic (-COO^−^) when it releases the proton in an aqueous solution. If a crystal of palmitic acid is placed on the surface of water in a beaker, not much will happen, but if it is warmed to approximately 63 °C, the crystal will melt and suddenly spread to form a monomolecular layer that fills the available surface area. At the molecular level, the hydrophilic carboxyl head group is interacting with the aqueous phase, while the hydrophobic hydrocarbon chains are standing vertically, each occupying around 0.2 nm^2^ of surface area. A measurement of surface tension will show an immediate decrease from 72 mN/m of water to approximately 57 mN/m. The difference between these two values—15 mN/m—is referred to as the surface pressure.

Now we can consider an example demonstrating how the surfactant effect can resolve a complex system. The Murchison meteorite is a mixture of silicate mineral grains with approximately 0.1% by weight of soluble organic compounds that include monocarboxylic acids, amino acids, alcohols, polycyclic aromatics such as pyrene and fluoranthene and even some purines such as adenine. Figure 2 shows a result reported by Mautner et al. [13] in which 20 mg of Murchison powder in 2 mL aqueous buffer was heated briefly to 100 °C at low and high pH ranges. The initial surface pressure was near zero mN/m at room temperature but increased dramatically to 8.1 mN/m when the powder was heated at pH 2, then increased further to 10.3 mN/m when heated at pH 11. The heat allowed small amounts of surface-active compounds to escape from the mineral powder and form a monolayer at the air–water interface. This represents a purification of the amphiphilic compounds by simply heating the original mixture in an aqueous solution. When the surface was cleaned, the surface pressure decreased but then rapidly increased as more surfactant molecules migrated to the interface and formed a monolayer.

Why might the assembly of surfactant monolayers and oils at air–water interfaces be significant for the origin of life? The reason is that ultraviolet light was likely to be an important energy source driving photochemical reactions, and polycyclic aromatic hydrocarbons (PAH) are pigments that are elevated to excited states when they absorb UV photons. Oils and amphiphiles that accumulate on aqueous surfaces would be exposed to UV light, and a variety of photochemical reactions would then occur. It has been proposed that such reactions could have served as a primitive version of photosynthesis involving electron transfer from a PAH such as pyrene to an acceptor such as benzophenone [14].

## 5. Bilayer Membranes Form Compartments by the Self-Assembly of Amphiphilic Compounds

Figure 3 illustrates how an increasing concentration can drive the self-assembly of amphiphilic compounds into membranous compartments. The amphiphile is decanoic acid, a monocarboxylic acid with a 9-carbon hydrocarbon chain and a carboxyl head group. In a dilute solution, the acid is present as individual molecules dissolved in an aqueous phase (A). If the concentration increases, for instance by evaporation, at some point the critical micelle concentration (CMC) is exceeded, and micelles begin to form (B). When the critical vesicle concentration (CVC) is exceeded, the micelles fuse into vesicles bounded by bilayers (C). At the highest concentration near dryness, the vesicles fuse into a multilamellar matrix (D), but then reassemble into vesicles upon rehydration (E).

This is another example of orderly structures emerging from disorderly, messy conditions. Membrane-bounded compartments were essential prerequisites for the origin of cellular life, and it seems likely that micelles and vesicles were abundant in the prebiotic environment [15]. Assuming that various polymers were also being synthesized, the microscopic membranous compartments would be able to encapsulate random peptides and oligonucleotides to generate protocells. Although protocells are not alive in the usual sense, they represent an essential step toward life. Each protocell differs in composition from all the rest and represents a microscopic experiment. Most are inert, and their components would be recycled, but a few might happen to have properties that allowed them to survive stresses imposed by the environment. Populations of protocells would therefore undergo the first stages of selection and evolution [12,16].

## 6. Adsorption as an Organizing Factor

Many physical and chemical processes cannot occur in the absence of a threshold concentration; so, a physical process that specifically increases the local concentration of a potential reactant can act as a catalyst. Clays are aluminum silicate minerals that have an enormous capacity to adsorb polar and ionic solutes. As shown in Figure 4, they are microscopic particles that are often present as layered structures called smectic phases. Various clays have surface areas ranging from 100 to 400 square meters per gram to which solutes can bind. James Ferris and co-workers tested whether clay surfaces could promote the polymerization of activated nucleotide monomers. In their initial studies, they reported that when the nucleotides saturated the surface of montmorillonite clay, oligomers ranging up to 10 nucleotides could be detected [17]. Later research demonstrated chain lengths as long as 30–50 nucleotides [18].

A chemist accustomed to reactions in aqueous solutions might be disappointed by the lack of polymerization in a solution of activated nucleotides, but then would be surprised to see polymers appear when the solution becomes more complex and “messier” with the addition of clay. The catalytic effect of clay minerals is an example of why messy conditions should be explored more extensively. There may be other surprises in store.

## 7. Eutectic Phases

The most common eutectic phase occurs when an aqueous solution freezes. As ice crystals form, they tend to exclude solutes, which become highly concentrated films between the crystals. The concentrating effect is similar to evaporation because reactions can occur that cannot proceed in a dilute, disordered solution. The power of eutectic phases to promote reactions was demonstrated by Monnard et al. [19] who froze complex mixtures of nucleotides at −18 °C. The nucleotides were chemically activated as imidazole esters and spontaneously formed ester bonds to polymerize into strands 5 to 14 nucleotides in length. Attwater et al. [20] also tested freezing temperatures as a way to promote the non-enzymatic polymerization of RNA. They used in vitro selection to evolve a cold-adapted ribozyme that was capable of catalyzing the template-directed synthesis of a 206-nucleotide RNA sequence. The reaction was markedly promoted by a cycle of freezing that concentrated the reactants in a eutectic phase.

Bada et al. [21] proposed that the early Earth may have passed through a “snowball” phase related to decreased solar luminosity at the time. They speculated that freeze–thaw cycles related to bolide impacts may have promoted reactions that synthesized organic compounds required for life’s origin. Eutectic phases associated with freezing would have played a role by concentrating reactants such as HCN, NH_3_ and aldehydes that would then undergo Strecker synthesis to form amino acids. Although this idea remains speculative, it does illustrate how another physical process, freezing and thawing in this case, has the potential to resolve specific reactions in complex conditions.

## 8. Duplex Structures of Nucleic Acids—Stacking and Base Pairing

Although three-dimensional crystals are common, and crystallization often serves to purify specific compounds from complex mixtures, the assembly of compounds into one-dimensional linear crystals is less familiar. We tend to take it for granted that the double helix of DNA is primarily stabilized by Watson–Crick base pairing between the two strands, but in fact stacking of the base pairs is even more important in terms of stabilization. Guanine monophosphate (GMP) quadruplexes are examples of spontaneous stacking that produces a linear crystal [22]. Figure 5 shows how four guanines form a quadruplex stabilized by eight hydrogen bonds indicated as dashed lines.

Atomic force microscopy can be used to visualize the linear crystals produced when a solution of GMP and cytidine monophosphate (CMP) is concentrated by evaporation at room temperature on a freshly cleaved mica sheet (Figure 6A). The GMP stacks into long, linear crystals that exclude the CMP. The linear crystals are adsorbed to three axes of the mica crystal, which accounts for their obvious alignment along three axes at 60°. Significantly, if the solution of GMP and CMP is evaporated on mica at 80 °C to provide activation energy, long polymeric strands emerge from the mixture [23], presumably composed of GMP and CMP linked by ester bond synthesis (Figure 6B).

This is an example of how two physical processes can drive a significant chemical reaction. The first process is evaporation that concentrates potential reactants on a mica surface so that one of the reactants forms quadruplexes and linear crystals. The second process is heating the mixture to provide activation energy that drives a condensation reaction leading to polymerization. These are conditions that are rarely used in the laboratory but common in hydrothermal sites associated with volcanism.

## 9. Thermodynamics, Kinetics and Wet–Dry Cycles

An important test of any chemical reaction proposed to be relevant to prebiotic chemistry is whether it is feasible in terms of thermodynamic principles [24]. A simplified description of the primary thermodynamic principle is that the free energy (ΔG) available to drive a reaction is a function of the change in enthalpy (ΔH) and entropy (ΔS) terms: ΔG = ΔH − TΔS. If heat can be released by a reaction, and disorder increases, the Gibbs free energy (ΔG) is negative, and the reaction is spontaneous.

If one visits volcanic regions on today’s Earth, a striking feature is that precipitation constantly supplies water to hydrothermal sites. A well-known example is Yellowstone National Park, but similar sites can be found in Kamchatka, Russia, New Zealand and Iceland. We refer to these as prebiotic analogues because similar sites would have been abundant on the early Earth 4 billion years ago before life began. A characteristic of hydrothermal sites is that the pools and hot springs undergo continuous cycles of evaporation and rewetting. There are two important properties of the wet–dry cycles that go beyond traditional laboratory methods. First, as a solution of potential reactants is concentrated to dryness, it makes chemical free energy available through the favorable changes in enthalpy and entropy described earlier. The enthalpy change is due to the reduction in water activity which draws the equilibrium of a condensation reaction to the right. The entropy change is favorable because as the reactants in a dilute solution become concentrated, they have an increased probability of interacting and losing water to the environment.

Finally, in the laboratory a typical reaction undergoes a single cycle in which reactions proceed downhill to products. However, multiple cycles available in natural conditions can take advantage of kinetic traps in which a forward reaction can be fast, but a back reaction is slow. For instance, the synthesis of ester bonds during the dry phase of a cycle can be very fast, but the spontaneous hydrolysis of polymers is slow. As a result, polymers accumulate in a steady state away from the thermodynamic equilibrium [24,25].

## 10. Stepping Out of the Laboratory into the Wild

A given reaction can work under carefully controlled laboratory conditions using pure reagents, buffered solutions at a specific pH and ionic solutes such as Mg^2+^ that may be required, but how valid is the assumption that it would have also been able to proceed on the prebiotic Earth? There is one way to become more confident of this assumption, which is to test whether a significant experiment also works in a prebiotic analogue site on today’s Earth. If the reaction is sufficiently robust, it will be reproduced even though such analogue sites are, by definition, messy and complex.

A few preliminary studies have been undertaken, and it is worth describing them here so that others might accept the challenge. Given that liquid water is essential for the origin of life, we should first note that on the Earth today there are two types of water that have been proposed as sites for the origin of life. Over 99% of the Earth’s water is salty seawater with 580 mM NaCl, 53 mM MgCl_2_ and 10 mM CaCl_2_; therefore, a common assumption is that life must have begun in the ocean. The assumption was supported when hydrothermal vents called “black smokers” were discovered, in which abundant microbial life used the reducing power of hydrogen sulfide (H_2_S) as a source of chemical energy [7]. The second type consists of the alkaline vents described earlier that emit a strongly alkaline solution containing up to 10 mM hydrogen gas in solution, which may serve as a source of reducing power [26].

The other 1% of the Earth’s water is distilled from salty seawater and falls as precipitation on subaerial continental land masses and volcanic islands like Hawaii and Iceland. Today, most of the freshwater is in the form of ice covering Antarctica and Greenland, but 4 billion years ago such extensive ice would have been absent when the global temperatures were estimated to be 55–80 °C. However, active volcanoes were likely to have been emerging from the ocean, and distilled water falling as precipitation would become incorporated into hydrothermal hot spring sites resembling those we see today.

Freshwater hot spring sites are an alternative to salty seawater for several reasons:The concentration of soluble cations is very low, in the range of a few millimolar [27].Because of sulfur dioxide emissions that dissolve in the water, the solution is in the acidic range, with pH 2–4.Cycles of evaporation and rewetting serve to concentrate potential reactants as films on mineral surfaces.

Hydrothermal fields such as those illustrated in Figure 1 are accessible prebiotic analogue sites which can be used to test the assumption that laboratory simulations represent processes and reactions expected to occur 4 billion years ago. A few such tests have been performed. For instance, Milshteyn et al. [28] asked whether membrane-bounded vesicles could assemble in hot spring conditions. Water samples were taken from Yellowstone hot spring pools at acidic and neutral pH ranges. Because contemporary membrane lipids such as phospholipid and cholesterol are products of metabolism, neither would have been available on the prebiotic Earth. Therefore a 1:1 mole ratio mixture of 12-carbon lauric acid and its monoglyceride was used to model the kinds of amphiphiles likely to be present. It was observed that vesicles readily formed in the hot spring water but not in seawater (Figure 7) because divalent cations of Ca^2+^ and Mg^2+^ caused the fatty acid to crystallize.

Similar experiments were performed by Joshi et al. [29] using a mixture of 40 mM decanoic acid and 20 mM monoglyceride exposed to hot spring water samples from the Puga region of Ladahk, India. Vesicles ranging from 5 to 10 μm in diameter were abundant, so it seems reasonable to assume that membranous compartments were present in freshwater hot springs on volcanic land masses emerging from the global ocean 4 billion years ago.

## 11. Conclusions

It is worth keeping in mind that a living cell is the messiest environment we can imagine, yet somehow, the biochemical reactions of metabolism proceed smoothly despite the incredible complexity of the cytoplasm. This is possible because enzymes catalyze specific reactions that allow those reactions to follow metabolic pathways established by several billion years of evolution. A second reason is that biophysical forces arrange structures within cells to optimize those functions. A prominent example is the self-assembly of biological membranes involved in electron transport and ATP synthesis. Although the kinds of lipids composing the membranes are under genetic control, the assembly process is not.

Another point is that the reaction of a pure, well-defined single reactant can generate what most researchers would call messy products. A classic example is the formose reaction in which hundreds of compounds are produced when formaldehyde (CHCO) reacts with itself in alkaline solutions. Another is the spontaneous polymerization of alpha hydroxy acids [9]. For instance, glycolic acid exposed to drying at 80 °C forms oligomers up to 12 subunits in length, with most between 3 to 6.

Although it is generally a good idea to use pure reagents and controlled conditions in the laboratory, it is also useful to realize that if we limit ourselves to the laboratory, we might miss an important ingredient or physical process. Which laboratory simulations of reactions related to the origin of life could be expected to proceed in prebiotic analogue conditions? This is a challenge to origins of life research because few investigators have been bold enough to perform such experiments in the field. The factors described here—concentration, crystallization, self-assembly of monolayers and bilayers, adsorption, eutectic phases, stacking and base pairing—can all occur in prebiotic environments and have the potential to promote desired reactions even in messy, complex conditions. Some of them, either singly or in combination, may provide significant clues to understand prebiotic chemistry that are not obvious in laboratory simulations.

## Figures and Tables

**Figure 1 life-12-01429-f001:**
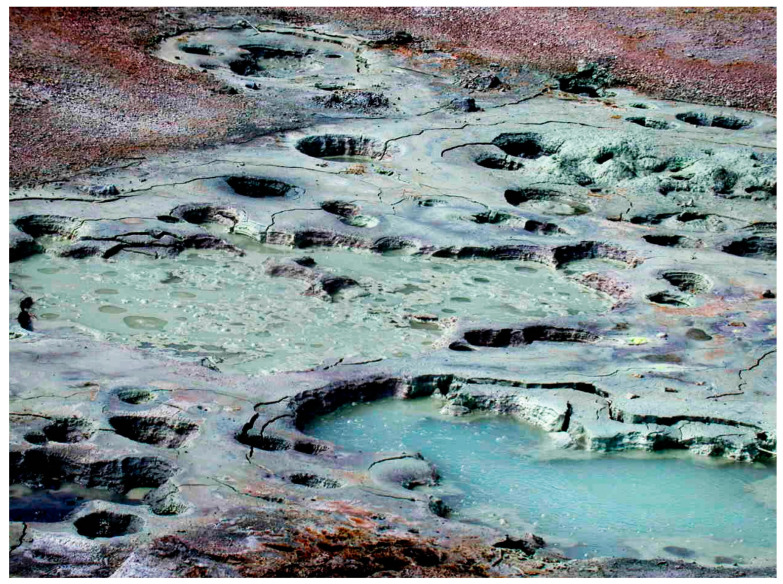
A messy, complex environment: Bumpass Hell on Mount Lassen, CA, USA. Credit: author.

**Figure 2 life-12-01429-f002:**
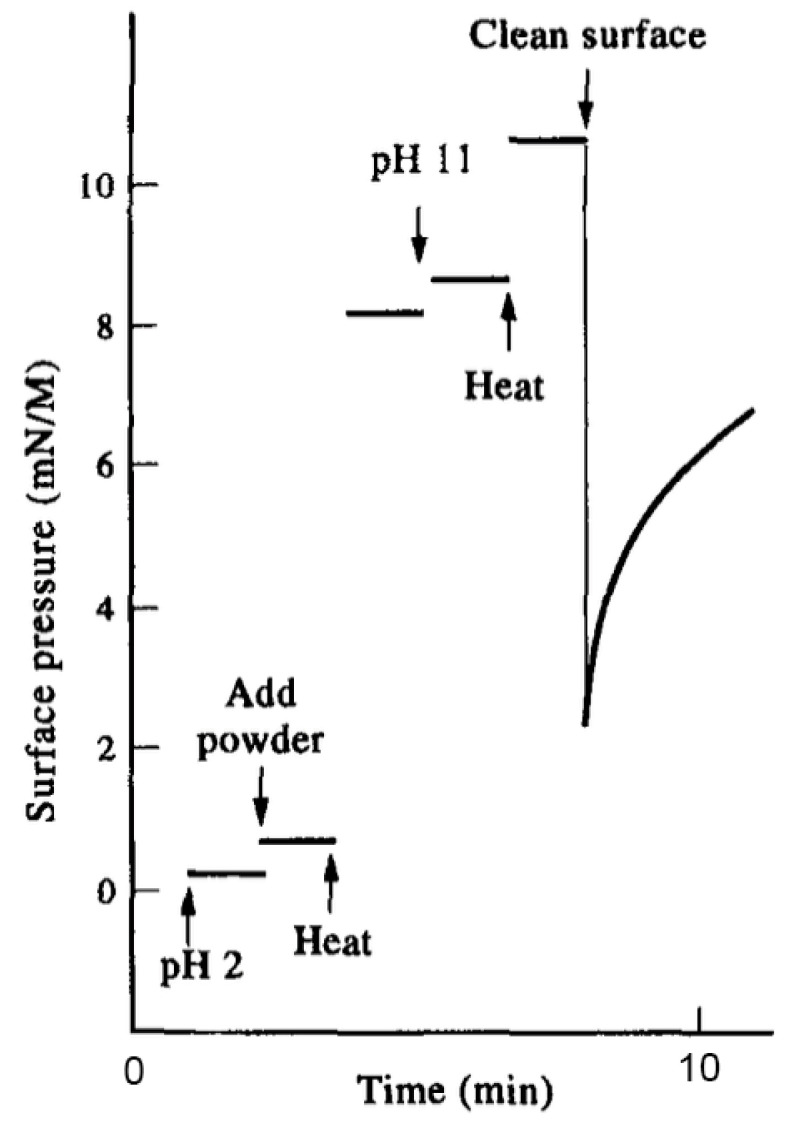
Monolayer assembly of amphiphilic compounds in the Murchison meteorite [13].

**Figure 3 life-12-01429-f003:**
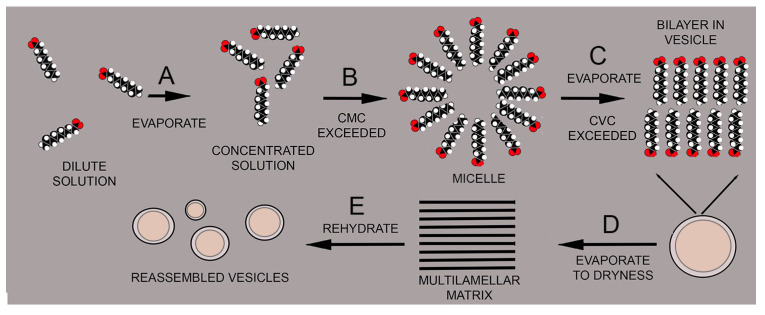
Self-assembly processes in solutions of amphiphilic molecules such as fatty acids.

**Figure 4 life-12-01429-f004:**
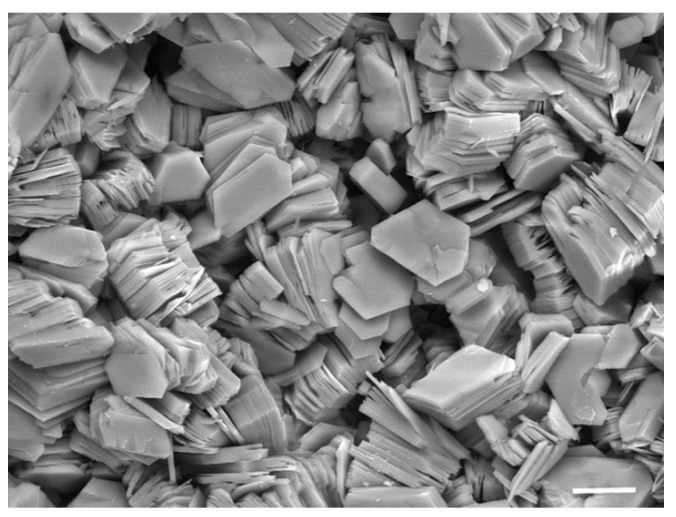
Scanning electron micrograph of clay crystals. The bar corresponds to 10 μm, equivalent to the diameter of a human red blood cell. Photograph courtesy of Evelyne Delbos, the James Hutton Institute.

**Figure 5 life-12-01429-f005:**
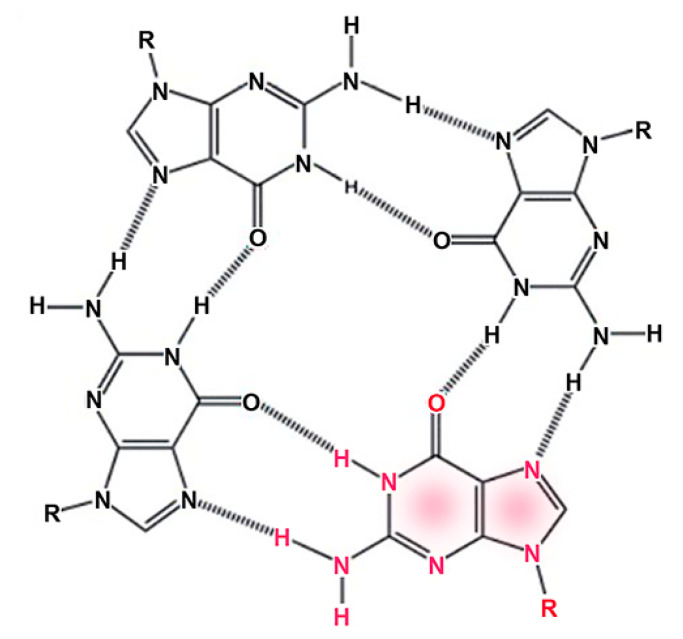
Guanine monophosphate assembles into a quadruplex structure stabilized by hydrogen bonding, indicated by dashed lines. The R groups represent the ribose phosphates linked to the guanine by N-glycoside bonds. One of the guanines is indicated by red fonts.

**Figure 6 life-12-01429-f006:**
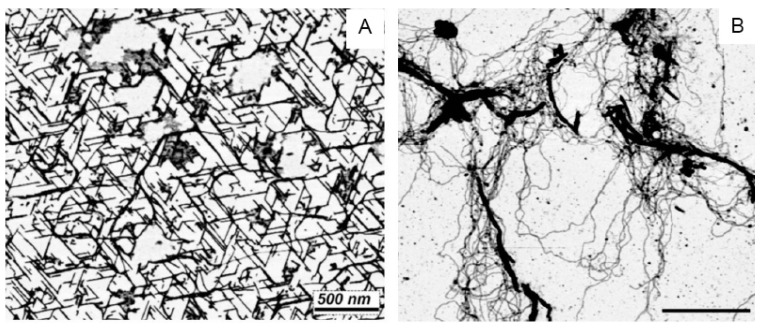
Linear strands of stacked guanosine monophosphate quadruplexes are shown in (**A**). The strands assemble when a mixture of GMP and CMP is dried on a freshly cleaved mica surface at room temperature, but CMP is excluded when the GMP forms quadruplexes aligned with the crystalline surface of the mica substrate. A different result occurs if the mixture is exposed to three wet–dry cycles at 80 °C (**B**). Instead of linear crystals, long strands of apparent polymers emerge, presumably composed of GMP and CMP linked by ester bonds [23].

**Figure 7 life-12-01429-f007:**
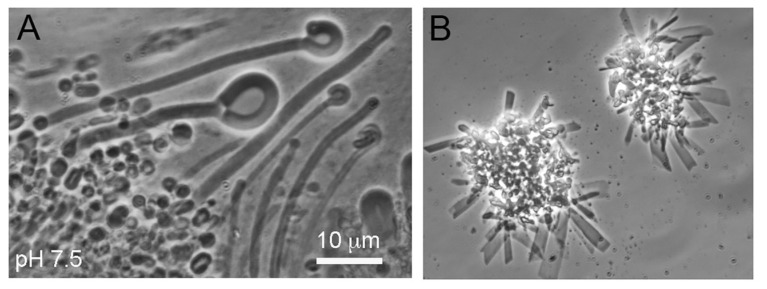
Membranous compartments assemble from a mixture of lauric acid and its monoglyceride exposed to hot spring water at pH 7.5 (**A**), but not in seawater at pH 8.1 (**B**). Image from Milshteyn et al. [28].

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
