# Peer review of "Origins of Life Research: The Conundrum between Laboratory and Field Simulations of Messy Environments"

_life, 2022, doi:10.3390/life12091429_

Round 1
Reviewer 1 Report
The presented article raises a fundamental problem for the search of the origins of life on Earth. It details with rigor the chemical criteria that require the researcher to leave the laboratory and go to Earth, in Nature, where life appeared 4 billion years ago. Even if the geochemical context has evolved during geological events, it is fundamental for all present and past life.
The "prebiotic" geochemical sites are numerous (the author cites several, New Zealand, Iceland ...) and we must find on these sites the samples and molecules that we can analyze in the laboratory and confront them with simulations that mimic the early days context of the primitive Earth.
Cycles of dryness and hydration, freezing and thawing, variable temperatures, ionic gradients etc. are evoked in this article which I recommend without reservation.
Author Response
I thank the reviewer for kind remarks.
No revision was suggested.
Reviewer 2 Report
Dear Author
I strongly reccomend the publication of this review
please follow the correction of the few suggestion directly noted into the manuscript copy in attachment

Author Response
I thank the reviewer for kind remarks.
The reviewer made a few suggestions in the text of the manuscript which I accepted and corrected.
Reviewer 3 Report
The author brings up a topic that is already a point of interest for many OoL workers. He adds that doing prebiotic chemistry in the “wild” is another option that prebiotic chemistry should consider. I agree with the statement and do think that this manuscript will add value to the present literature. However, I do have some major concerns about how the author came to his major discussion points. Below is a list of comments that I highly recommended the author add/amend in his manuscript.
---
-
The title needs some changes to be a bit more informative. Not doing so will allow the paper to go under the radar. I definitely will not go with the present title. Perhaps this title might work: What every prebiotic chemist should know: the conundrum between laboratory and field simulations of messy environments to study the origins of life. This is just a suggestion, but I’m open to a better title.
-
In the introduction, the author briefly mentioned what “messiness” means, but didn’t elaborate enough, and give adequate examples, considering the much new literature that has spoken extensively on messiness. For example, one meaning of messiness is that the word is referring to the chemical space of prebiotic chemistry that includes many non-biological and some biological compounds in specific OoL sites (Miller-Uray, Carbonaceous Chondrite, etc). We have many examples of this, for example, Schmitt Koplin 2010, Gutenberg, 2017, Wolos, 2020, etc. I would recommend the author include these references.
-
As for the term “messy conditions” mentioned in the manuscript. I notice that the author is clumping both prebiotic chemical space and (any) prebiotic geochemical condition/environment. Note that messy chemistry was coined in Gutenberg's work, and I would urge the author to segregate the chemical space part and the fluctuating physical conditions of the geochemical location to study the OoL. Or use another term to include both criteria. Using “messy” for both simply confuse the reader.
-
In line 56, the term “tar” is used. The author should elaborate on its meaning and do cite Schwartz, 2007 paper entitled “Intractable mixtures and the origin of life”, or any relevant paper addressing tar.
-
The author gives ester as an example in lines 86 - 90, but didn’t give any reference to prebiotic polyester formations (e.g., Chandru et al 2018, 2021, etc), which is an active field of interest when comes to messy prebiotic chemistry and using non-biological compounds as a tool to study the OoL. The author also mentions wet/dry synthesis but didn’t list his own paper as a reference (e.g., Deamer and Ross, 2016). Since this is a review paper, an extensive list of references is warranted.
-
In lines 172 -174, the author should include examples from Sean Jordon, 2019, and Susovan Sarkar, 2020 since the differing composition of protocells is discussed in those papers and is very relevant to the author’s own manuscript.
Author Response
I thank the referee for knowledgeable comments that improved the manuscript.
- The title needs some changes to be a bit more informative
The title has been clarified: Origins of life research: The conundrum between laboratory and field simulations of messy environments
- In the introduction, the author briefly mentioned what “messiness” means, but didn’t elaborate enough, and give adequate examples, considering the much new literature that has spoken extensively on messiness. For example, one meaning of messiness is that the word is referring to the chemical space of prebiotic chemistry that includes many non-biological and some biological compounds in specific OoL sites (Miller-Uray, Carbonaceous Chondrite, etc). We have many examples of this, for example, Schmitt Koplin 2010, Gutenberg, 2017, Wolos, 2020, etc. I would recommend the author include these references.
The description of “messiness” has been expanded and the suggested references are now cited.
- As for the term “messy conditions” mentioned in the manuscript. I notice that the author is clumping both prebiotic chemical space and (any) prebiotic geochemical condition/environment. Note that messy chemistry was coined in Gutenberg's work, and I would urge the author to segregate the chemical space part and the fluctuating physical conditions of the geochemical location to study the OoL. Or use another term to include both criteria. Using “messy” for both simply confuse the reader.
I agree and have now distinguished between messy chemistry and fluctuating physical conditions. I pointed out that the two can have a synergistic interaction in which the physical environment guides specific chemical reactions.
- In line 56, the term “tar” is used. The author should elaborate on its meaning and cite Schwartz, 2007 paper entitled “Intractable mixtures and the origin of life”, or any relevant paper addressing tar.
I expanded the definition of tar and now cite Schwartz 2007 and Benner et al. 2012.
- The author gives ester as an example in lines 86 - 90, but didn’t give any reference to prebiotic polyester formations (e.g., Chandru et al 2018, 2021, etc), which is an active field of interest when comes to messy prebiotic chemistry and using non-biological compounds as a tool to study the OoL. The author also mentions wet/dry synthesis but didn’t list his own paper as a reference (e.g., Deamer and Ross, 2016). Since this is a review paper, an extensive list of references is warranted.
Chandru et al is now cited as well as two additional reverences that are pertinent. Ross and Deamer 2016 was already cited.
- In lines 172 -174, the author should include examples from Sean Jordon, 2019, and Susovan Sarkar, 2020 since the differing composition of protocells is discussed in those papers and is very relevant to the author’s own manuscript.
I have added the Sarkar citation as suggested and a discussion of differing composition of protocells. I was unable to find the Sean Jordon reference in the literature.
Round 2
Reviewer 3 Report
The author responded well to the queries and suggestions.